# A Comprehensive Benchmark of Supervised and Self-supervised Pre-training on Multi-view Chest X-ray Classification

**Muhammad Muneeb Afzal**[*1]                                    MUNEEB.AFZAL@NYU.EDU
**Muhammad Osama Khan**[*1]                                    OSAMA.KHAN@NYU.EDU
**Yi Fang**[1,2]                                                              YFANG@NYU.EDU
[1]*New York University, New York, USA*
[2]*New York University Abu Dhabi, Abu Dhabi, UAE*

**Editors:** Accepted for publication at MIDL 2024

## Abstract

Chest X-ray analysis in medical imaging has largely focused on single-view methods. However, recent advancements have led to the development of multi-view approaches that harness the potential of multiple views for the same patient. Although these methods have shown improvements, it is especially difficult to collect large multi-view labeled datasets owing to the prohibitive annotation costs and acquisition times. Hence, it is crucial to address the multi-view setting in the low data regime. Pre-training is a critical component to ensure efficient performance in this low data regime, as evidenced by its improvements in natural and medical imaging. However, in the multi-view setup, such pre-training strategies have received relatively little attention and ImageNet initialization remains largely the norm. We bridge this research gap by conducting an extensive benchmarking study illustrating the efficacy of 10 strong supervised and self-supervised models pre-trained on both natural and medical images for multi-view chest X-ray classification. We further examine the performance in the low data regime by training these methods on 1%, 10%, and 100% fractions of the training set. Our best models yield significant improvements compared to existing state-of-the-art multi-view approaches, outperforming them by as much as 9.9%, 8.8% and 1.6% on the 1%, 10%, and 100% data fractions respectively. We hope this benchmark will spur the development of stronger multi-view medical imaging models, similar to the role of such benchmarks in other computer vision and medical imaging domains. As open science, we make our code publicly available to aid in the development of stronger multi-view models.

**Keywords:** Multi-view X-ray, self-supervised, pre-training.

## 1. Introduction

Medical imaging analysis has benefited significantly from improvements in deep learning and computer vision in recent years (Rajpurkar et al., 2020, 2021). These developments have been largely enabled by the availability of large chest X-ray datasets. However, most existing methods (Rajpurkar et al., 2017, 2018; Li et al., 2018; Cohen et al., 2019) still largely rely on a single view (e.g. frontal) for chest X-ray analysis. In clinical practice, however, some structures and pathologies are more readily distinguishable from a lateral X-ray (Bertrand

---

et al., 2019). Hence, it is particularly appealing to develop methods that utilize multiple views, such as frontal (PA) and lateral (L) views. To this end, recently methods have been developed that exploit multiple views to improve chest X-ray recognition. For instance, DualNet (Rubin et al., 2018) observed an improvement of 3% mean AUC by utilizing both PA and L views. Similarly, Hashir et al. (Hashir et al., 2020) studied various methods of combining PA and L views, resulting in improvements on 32 pathologies.

However, due to the challenges associated with obtaining large labeled datasets for medical imaging in general and multi-view imaging in particular, it is especially pertinent to focus on the low data regime in the multi-view setting. In such low data regimes, transfer learning via either supervised or self-supervised pre-training has shown significant improvements in both natural imaging (Xu et al., 2022; He et al., 2020; Chen et al., 2020b) and medical imaging (Azizi et al., 2022) settings. For instance, Taher et al. (Hosseinzadeh Taher et al., 2021) recently presented a thorough study illustrating the efficacy of various supervised and self-supervised learning methods for single-view medical imaging. However, in the multi-view setup, such pre-training strategies have received relatively little attention and ImageNet initialization remains largely the norm.

We bridge this gap by providing an extensive study of various pre-training methods for multi-view chest X-ray classification in the low data regime. While the efficacy of supervised and self-supervised pre-training in medical imaging is known, our work addresses the research gap of its applicability in the multi-view setting. Our focus on low-data regimes, pertinent due to the high creation costs of multi-view datasets, highlights our study's practical value, offering insights that are immediately relevant to current healthcare settings. Specifically, we investigate the low data regime by training on 1%, 10%, and 100% fractions of the training set. We evaluate strong supervised and self-supervised learning methods that have demonstrated significant performance enhancements across a range of domains, including computer vision and medical imaging.

Concretely, we study four transfer learning strategies (Fig. 1) – 1) supervised learning on natural images, 2) self-supervised learning on natural images, 3) supervised learning on medical images, and 4) self-supervised learning on medical images. We fine-tune these pre-trained models for multi-view chest X-ray classification on the PadChest dataset and achieve improvements of as much as 9.9% in the low data regime compared to existing state-of-the-art methods. To summarize, we provide a timely benchmarking study investigating several strong pre-training methods for multi-view chest X-ray recognition.

Briefly, our main contributions are:

- We present the first extensive multi-view benchmarking study illustrating the efficacy of 10 strong supervised and self-supervised methods pre-trained on both natural as well as medical images.

- We are the first to systematically study multi-view chest X-ray classification in a low-data regime by training on various different data fractions. This data-efficient setting is of significant practical importance owing to the huge data acquisition times and annotation costs associated with collecting large *multi-view* labeled datasets.

- Our best models yield significant improvements compared to existing state-of-the-art multi-view approaches, outperforming them by as much as 9.9%, 8.8% and 1.6% on the 1%, 10% and 100% data fractions respectively.

## 2. Related Work

Although chest radiography has been investigated extensively in the medical imaging community, most methods have relied on a single (i.e. the frontal) view. Many of these works (Yao et al., 2017; Rajpurkar et al., 2017; Guan et al., 2018; Kumar et al., 2018; Guan et al., 2018; Baltruschat et al., 2019) use the frontal posteroanterior (PA) view of chest x-rays to classify different diseases. Recently, however, multiple views (frontal and lateral) have also been used, emulating the usual radiology practice where multiple views are taken into account for diagnosis of chest X-rays. For instance, (Rubin et al., 2018) proposed DualNet that used two separate branches for frontal and lateral views to predict diseases. (Bertrand et al., 2019; Hashir et al., 2020) demonstrated that including lateral views with frontal views enhances performance on 32 PadChest labels. Building on DualNet, they introduced a revised architecture with auxiliary losses and curriculum learning.

Recently, (Hosseinzadeh Taher et al., 2021) presented a benchmarking analysis of several strong transfer learning techniques for medical imaging. However, in contrast to (Hosseinzadeh Taher et al., 2021) who investigate *single-view* tasks, we study these pre-training techniques for *multi-view* chest X-ray analysis since it is not clear a-priori that the same trends translate to the multi-view setting owing to the unique training dynamics of multi-view systems (Wu et al., 2020, 2022). Moreover, we focus on the data-efficient multi-view setting, which is of great clinical relevance but has not yet been explored. Hence, our work bridges these gaps by extensively evaluating various supervised and self-supervised pre-training strategies for *data-efficient multi-view* chest X-ray analysis. More details on related multi-view methods and pre-training methods can be found in Appendix A.

## 3. Methods

Fig. 1 shows an overview of our approach. In the first stage, we pre-train our networks using one of the following three approaches: 1) Supervised learning on natural images, (2) Self-supervised learning on natural images, or (3) Supervised learning on medical images and 4) Self-supervised learning on medical images. In the second stage, for downstream fine-tuning in the multi-view setting, we select the pre-trained backbone and use it to initialize both the frontal (PA) and lateral (L) view models. This multi-view model is then fine-tuned with the downstream data on the PadChest dataset.

**Implementation Details** We use the pre-trained backbones from the official implementations, using the code provided by (Hosseinzadeh Taher et al., 2021) to pre-process and load the models. All our experiments use a ResNet50 backbone except for self-supervised methods in the medical domain where densenet121 is the de-facto standard. We applied a standardized set of hyperparameters across models for comparable results. All the multi-view models are implemented in PyTorch and are trained for 100 epochs with an Adam optimizer, a batch size of 8, and a learning rate of $1e^{-4}$ on an NVIDIA V100 GPU.

### 3.1. Dataset

In the past, most publicly accessible chest X-ray datasets only offered one view (frontal) (Rajpurkar et al., 2017, 2018; Li et al., 2018; Cohen et al., 2019). However, recently, with the availability of large chest X-ray datasets (Irvin et al., 2019; Johnson et al., 2019; Bustos et al.,

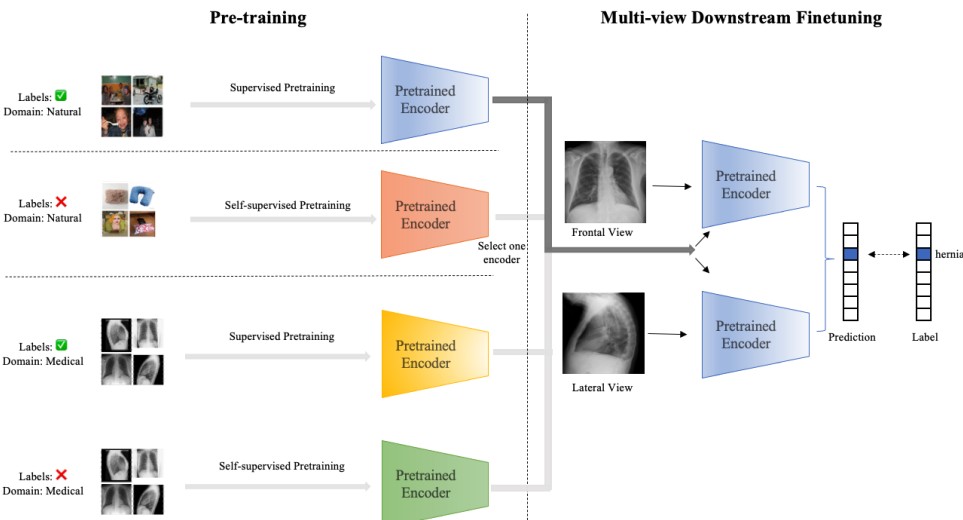

Figure 1: **Left**: Four pre-training strategies – 1) Supervised learning on natural images, 2) Self-supervised learning on natural images, 3) Supervised learning on medical images and 4) Self-supervised learning on medical images. **Right**: Supervised fine-tuning with pre-trained encoders on paired frontal and lateral chest X-rays.

2020) with paired frontal and lateral views, it has become feasible to investigate the efficacy of paired multi-view data for chest X-ray analysis. For this study, the PadChest dataset (Bustos et al., 2020), which contains 160,868 chest x-ray images including paired frontal (PA) and lateral views, is utilized. We selected PadChest for our benchmarking effort due to its comprehensive annotations and diverse multi-view chest X-ray images. We prepare the multi-view dataset using the same setup as that followed by (Hashir et al., 2020). Distinct from prior multi-view models, we introduce a method to evaluate the efficacy of our models on different training data fractions. To this end, we devise the following three fractions of the train set of the PadChest multi-view dataset described above: 1%, 10%, and 100%. In creating the 1% and 10% data fractions, we aimed to maintain a distribution of conditions that closely mirrors the full dataset. This process involved random sampling of images to roughly preserve the pathology distribution for each subset. This setup helps us evaluate how decreasing dataset size affects multi-view chest X-ray classification performance. For a fair comparison with prior works (Hashir et al., 2020), we use the same 60/20/20 split for train, validation and test respectively, and use the AUC (Area under the ROC) metric for evaluation. We conduct each experiment three times, with each run representing a different randomly selected training split to obtain more accurate results.

### 3.2. Pre-training

For pre-training, we experiment with 10 different methods (see Fig. 2) which can be categorized into the following: 1) supervised learning on natural images, 2) self-supervised learning on natural images, 3) supervised learning on medical images, and 4) self-supervised learning on medical images.

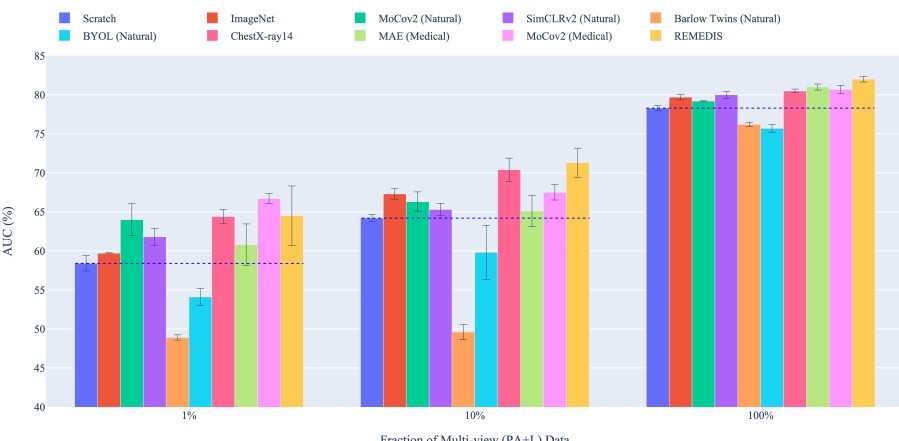

Figure 2: Mean AUC of models fine-tuned on 1%, 10%, and 100% data fractions across 64 conditions compared: 5 models pre-trained on natural images and 4 on medical images versus a random baseline (Scratch). Pre-training methods outperform Scratch, especially in data-efficient settings (1% and 10% fractions).

*Scratch* refers to the baseline method, where we train the multi-view model starting from a random initialization. For the first category, for supervised learning on natural images, we use the standard ImageNet dataset (Deng et al., 2009). Secondly, we employ the following four strong self-supervised learning methods on natural images: MoCov2 (Chen et al., 2020c), SimCLRv2 (Chen et al., 2020b), BYOL (Grill et al., 2020), and Barlow Twins (Zbontar et al., 2021). These methods are representative of major seminal works in the self-supervised domain. Thirdly, for supervised pre-training on medical images, we leverage the Chest X-ray14 (Wang et al., 2017) which is the de-facto dataset for Chest X-ray classification. Lastly, for self-supervised learning on the medical domain, we experiment with MoCo-v2 (Chen et al., 2020c) and Masked Auto Encoder (MAE) (He et al., 2022) that are pre-trained on medical data in a self-supervised fashion (Xiao et al., 2023). Furthermore, we also experiment with Robust and Efficient Medical Imaging with Self-Supervision (REMEDIS) (Azizi et al., 2022) which is the state-of-the-art method in self-supervised medical imaging that combines supervised transfer learning with self-supervised learning.

### 3.3. Multi-view Downstream Fine-tuning

For the downstream task, we use the pre-trained encoder (Fig. 1) and use it to initialize both the frontal and lateral backbones, which are shared during training. The representations of both these backbones are combined before finally yielding the classification prediction. This multi-view network is trained and evaluated on the PadChest dataset, containing paired frontal and lateral views, described above.

## 4. Results

In this section, we present extensive results and analyses illustrating the efficacy of various pre-training methods for multi-view chest X-ray analysis. Concretely, we study supervised

Table 1: Mean AUC for models fine-tuned on 1%, 10%, and 100% of multi-view labeled data across 64 conditions, comparing 10 methods against a random initialization baseline. Pre-training methods generally outperform the baseline, especially in low data regimes (1% and 10% data fractions).

| Domain | Type | Method | AUC (1%) | AUC (10%) | AUC (100%) |
|---|---|---|---|---|---|
| - | - | Scratch | $58.4 \pm 0.98$ | $64.2 \pm 0.48$ | $78.3 \pm 0.35$ |
| Natural | Supervised | ImageNet | $59.7 \pm 0.06$ | $67.3 \pm 0.67$ | $79.7 \pm 0.34$ |
| | Self-supervised | MoCov2 | $64.0 \pm 2.06$ | $66.3 \pm 1.26$ | $79.2 \pm 0.06$ |
| | | SimCLRv2 | $61.8 \pm 1.09$ | $65.3 \pm 0.79$ | $80.0 \pm 0.45$ |
| | | Barlow Twins | $48.9 \pm 0.34$ | $49.6 \pm 0.971$ | $76.2 \pm 0.28$ |
| | | BYOL | $54.1 \pm 1.07$ | $59.8 \pm 3.45$ | $75.7 \pm 0.48$ |
| Medical | Supervised | ChestX-ray14 | $64.4 \pm 0.92$ | $70.4 \pm 1.49$ | $80.5 \pm 0.24$ |
| | Self-supervised | MAE | $60.8 \pm 2.66$ | $65.1 \pm 1.99$ | $81.0 \pm 0.39$ |
| | | MoCov2 | $\mathbf{66.7 \pm 0.65}$ | $67.5 \pm 0.98$ | $80.7 \pm 0.53$ |
| | | REMEDIS | $64.5 \pm 3.82$ | $\mathbf{71.3 \pm 1.87}$ | $\mathbf{82.0 \pm 0.35}$ |

and self-supervised learning methods trained on both natural and medical imaging domains. Sec. 4.1 presents the benchmark of 10 different pre-training methods for data-efficient multi-view learning. This is followed by a fine-grained analysis of the effect of different pre-training strategies on multi-view classification in Sec. 4.2. Lastly, Sec. 4.3 compares the performance of these pre-trained models against current SOTA multi-view methods, where it surpasses them by significant margins, especially in the low data regime. We further delve into the clinical effectiveness of the aforementioned pre-training methods, conducting a detailed analysis on the 14 commonly occuring diseases featured in the ChestXray14 dataset (see Appendix C).

## 4.1. Multi-view Chest X-ray Classification Benchmark

Here, we provide a thorough benchmark of 10 different methods, including 5 pre-trained on natural images and 4 pre-trained on medical images, as illustrated in Fig. 2. For completeness, we compare all the methods against a random initialization baseline. Within both the natural and medical imaging pre-training settings, 1 method is pre-trained via supervised learning whereas others are pre-trained via self-supervised learning owing to the superiority of self-supervised learning witnessed in recent years. Table 1 shows the mean AUC (on 64 pathologies) across 3 labeled fractions (1%, 10% and 100%) of the multi-view PadChest dataset (Bustos et al., 2020).

Observe that 100% here corresponds to ∼19k labeled pairs of (PA+L) views which is especially difficult to collect, requiring years of acquisition time and hundreds of clinical annotation hours. This issue is exacerbated for *multi-view* data acquisition and annotation. Hence, it is pertinent to focus on the data-efficient fractions in the *multi-view* setting.

Table 1 demonstrates the significant gains achieved by leveraging supervised and self-supervised pre-training for multi-view chest X-ray diagnosis. Except for Barlow Twins and BYOL, all of the pre-training methods surpass the random initialization baseline used in prior works (Rubin et al., 2018; Hashir et al., 2020) across all 3 data fractions, highlighting the efficacy of these pre-training methods. The underperformance of Barlow Twins and

BYOL suggests their optimization strategies might not be fully compatible with the intricacies of multi-view chest X-ray data. Specifically, the Barlow Twins' focus on reducing feature redundancy could remove essential, nuanced details vital for accurate X-ray classification. Similarly, BYOL relies on an asymmetrical architecture where the online network predicts the target network's representation. This setup might not effectively capture the nuanced differences between multiple views of chest X-rays. The similarity in performance between supervised pre-training with ChestXray14 and self-supervised pre-training with REMEDIS is notable and highlights the advantage of domain-specific pre-training. However, it is crucial to note that such direct, condition-specific supervision is often unfeasible in real-world applications. Hence, in cases where such large labeled datasets are not readily available, effective SSL methods like REMEDIS are pivotal.

The advantages of pre-training methods become evident in the low data regime. For instance, in the data-efficient setting of 1% and 10%, the best methods, MoCov2 (medical) and REMEDIS, surpass the random baseline by big margins of 8.3% and 7.1% respectively. Hence, self-supervised and supervised pre-training is highly effective for multi-view chest X-ray diagnosis. Given these significantly improved performances, we hope practitioners will switch to using these pre-trained models as an alternative to random initialization.

### 4.2. Comparative Analysis of Pre-trained Models

To decouple the effects of different pre-training strategies and datasets on downstream multi-view chest X-ray diagnosis, we compare the random initialization baseline against the best performing models from the following pre-training strategies – 1) supervised learning on natural images, 2) self-supervised learning on natural images, 3) supervised learning on medical images, and 4) self-supervised learning on medical images.

**Supervised or Self-supervised:** Overall, we observe that self-supervised pre-training tends to outperform supervised pre-training across both the natural and medical imaging domains (see Fig. 3). This trend aligns well with the recent works (Azizi et al., 2022; Chen et al., 2020b,c; Zbontar et al., 2021) in computer vision and medical imaging which show the superiority of self-supervised approaches over supervised pre-training.

**Natural or Medical:** Across both supervised and self-supervised methods, in-domain pre-training on medical images yields better representations for the downstream multi-view chest X-ray classification task, as evidenced by the results in Fig. 3. This highlights the efficacy of in-domain pre-training and emphasizes the importance of acquiring more data, especially in the multi-view setting in the medical imaging domain.

**Importance of Medical Pre-training:** To show that the performance gains in our study are primarily due to pre-training rather than the specific method used, we conduct an ablation study with SimCLR v2 (Chen et al., 2020b) and SimCLR v2 + Medical-Pretraining (REMEDIS) (Azizi et al., 2022). Table 2 demonstrates that the performance of REMEDIS is largely attributed to pre-training on medical data. Specifically, the REMEDIS framework, which builds upon SimCLR v2 by incorporating additional pre-training on various medical datasets, demonstrates marked improvements. These results indicate that pre-training on medical data leads to significant performance improvements. Furthermore, we conduct additional ablation experiments to explicitly delineate the benefits of pre-training with medical data. Detailed experimental setup and results are provided in Appendix B.

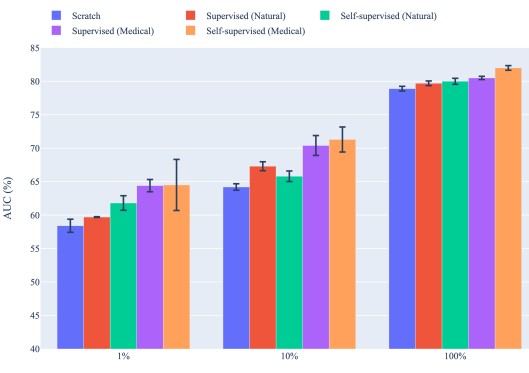

| Method | Pre-train | AUC (1%) | AUC (10%) | AUC (100%) |
|---|---|---|---|---|
| DualNet | ✗ | 52.7 | 62.4 | 80.1 |
| Hemis | ✗ | 48.8 | **62.5** | 80.3 |
| Stacked | ✗ | 54.0 | 61.7 | **80.4** |
| AuxLoss | ✗ | **54.6** | 62.2 | 80.3 |
| REMEDIS | ✓ | **64.5 ± 3.82** | **71.3 ± 1.87** | **82.0 ± 0.35** |

Figure 3: **Left:** Comparison of best-performing models across four categories. Self-supervised pre-training on medical data outperforms all methods, significantly exceeding the random baseline, notably in low data scenarios. **Right:** Comparison with SOTA multi-view chest X-ray classification methods across three data fractions (for 64 conditions). Our best pre-training model outperforms SOTA by huge margins of 9.9%, 8.8% and 1.6% on the 1%, 10% and 100% data fractions.

## 4.3. Comparison with SOTA Multi-view Methods

In this section, we compare our best performing method against state-of-the-art multi-view chest X-ray classification algorithms. As illustrated in Fig. 3, our best performing model *REMEDIS* outperforms the state-of-the-art methods by huge margins of 9.9%, 8.8% and 1.6% on the 1%, 10% and 100% data fractions. Perhaps even more strikingly, our best performing model does not leverage any of the specialized techniques employed by the other SOTA methods such as using auxiliary losses (Hashir et al., 2020) or combining multiple views by computing pixel-wise statistics of the two branches (Havaei et al., 2016).

## 5. Conclusion

Multi-view methods are more appealing compared to single-view methods since some pathologies are more readily identifiable from a particular view. However, owing to the challenges associated with collecting large multi-view labeled datasets, it is imperative to focus on the data-efficient setting when developing multi-view models. Pre-training is critical to yield good performance in such low data regimes. Hence, we present the first thorough benchmarking study illustrating the efficacy of 10 strong supervised and self-supervised models pre-trained on both natural and medical images for multi-view chest X-ray classification. Our best models yield significant improvements compared to existing state-of-the-art multi-view approaches, outperforming them by as much as 9.9%, 8.8% and 1.6% on the 1%, 10% and 100% data fractions respectively. Moreover, we observe that self-supervised learning on medical images, followed by supervised fine-tuning on single-view chest X-ray classification, provides the most optimal transfer learning strategy for multi-view chest X-ray analysis. We hope this benchmark will act as a catalyst for the development of stronger multi-view medical imaging models, analogous to the role of similar benchmarks in other domains of computer vision and medical imaging.

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

## Appendix A. Detailed Related Works

### A.1. Multi-view Methods

In medical imaging, multiple views have been employed for various tasks. For instance, (Setio et al., 2016) proposed a multi-view CNN architecture that fuses candidates from three different pulmonary nodules for detection. (Shachor et al., 2020) introduced Mixture of Views in which they combined different views for classification of breast microcalcifications. (Geras et al., 2017) proposed a multi-view deep CNN that takes in different views of each breast for breast cancer screening. (Carneiro et al., 2017) used unregistered multi-view mammograms for classification of breast cancer. MVMT (mult-view multi-task) (Kyono et al., 2019) presented a CNN architecture to predict patient features that are related with cancer. Similarly, (Khan et al., 2019) proposed Multi-View Feature Fusion that fused four views in order to classify mammograms. (Bermejo-Peláez et al., 2018) employed different views of pulmonary segment tissue for full lung classification. (Kitamura et al., 2019) utilized multiple views for ankle fracture detection. MVP-Net (Li et al., 2019) used a multi-view FPN (feature pyramid network) for lesion detection whereas (Lopez et al., 2022) learnt correlations between four views of mammograms using hypercomplex neural networks for cancer classification.

### A.2. Pre-training Methods

Self-supervised and supervised learning methods trained on both natural and medical imaging domains are generally used for pre-training downstream networks. Self-supervised Learning (SSL) is a type of representation learning method where unlabeled data is leveraged to learn meaningful representations. Following (Hosseinzadeh Taher et al., 2021), we analyze the several types of pre-training methods. One of the seminal works from these methods, SimCLR (Chen et al., 2020a) showed that composing different augmentations, using an extra non-linear projection head, and (3) using bigger batch sizes improves performance in the SSL domain. MoCo (He et al., 2020) bypassed the need for large batch sizes by introducing a novel dynamic dictionary that saves the encoded keys in a queue. BYOL (Grill et al., 2020) introduced a novel method where the online network tries to predict the representation of the target network when each network is given a different augmented input image. Unlike the previous methods, this does not require any negative pairs. Barlow Twins (Zbontar et al., 2021) outperformed the previous methods by proposing a simple mechanism in which the cross-correlation matrix of two embedded representations of the distorted images is calculated and is then made as close to the identity matrix. Masked Autoencoders (MAE) (He et al., 2022) is a self-supervised technique in which the input image is divided into patches, and the network is trained to predict the masked parts of the image. Particularly, the decoder component of the network is provided with the input containing the masked segments to reconstruct the original image, while the encoder is not fed with the masked parts. By training the network to predict the masked parts of the image, MAE essentially learns to extract meaningful representations from the image, which can then be used for downstream tasks. Specifically, for self-supervised learning in the medical domain, (Azizi et al., 2022) introduced Robust and Efficient Medical Imaging with Self-Supervision (REMEDIS), a strategy that improves the robustness and data efficiency

in medical imaging domain. By integrating large-scale supervised transfer learning with self-supervised learning, REMEDIS minimizes the need for task-specific adjustments.

For supervised pre-training on natural images, ImageNet pre-training (Deng et al., 2009) has been the de-facto standard. In the medical imaging domain, it is customary to employ pre-trained networks on the widely recognized ChestX-ray14 dataset (Wang et al., 2017) for conducting chest X-ray analysis.

In our work, by leveraging the previously mentioned prominent methods in supervised and self-supervised learning, we analyze the efficacy of 10 strong supervised and self-supervised models pre-trained on both natural and medical images for multi-view chest X-ray classification.

## Appendix B. Importance of Pre-training

Table 2 shows the importance of medical pre-training as indicated by the superior performance of *SimCLR v2 + Medical-Pretraining* (REMEDIS) compared to *SimCLR v2*.

Table 2: Ablation study showing the impact of pre-training on medical data.

| Method | AUC (1%) | AUC (10%) | AUC (100%) |
|---|---|---|---|
| Scratch | $60.7 \pm 1.02$ | $67.1 \pm 0.43$ | $81.2 \pm 0.837$ |
| SimCLR v2 | $65.3 \pm 1.67$ | $69.4 \pm 1.60$ | $81.3 \pm 0.381$ |
| SimCLR v2+Med-Pretrain (REMEDIS) | $69.0 \pm 3.67$ | $73.7 \pm 1.26$ | $84.3 \pm 2.39$ |

To further investigate the impact of fine-tuning on single-view medical data, we conduct additional experiments using the MoCo-v2 and MAE methods. These experiments aim to isolate the benefits of pre-training by comparing performance before and after fine-tuning on single-view medical images. The results of these experiments are shown in Table 3 below.

Table 3: Ablation study showing the impact of fine-tuning on single-view medical data.

| Method (Medical) | AUC (1%) | AUC (10%) | AUC (100%) |
|---|---|---|---|
| MoCo-v2 | $66.7 \pm 0.65$ | $67.5 \pm 0.98$ | $80.7 \pm 0.53$ |
| MoCov2-FT | $70.3 \pm 0.74$ | $69.2 \pm 0.78$ | $82.2 \pm 0.32$ |
| MAE | $60.8 \pm 2.66$ | $65.1 \pm 1.99$ | $81.0 \pm 0.39$ |
| MAE-FT | $70.0 \pm 0.42$ | $69.2 \pm 0.94$ | $81.5 \pm 0.18$ |

For this study, we first experiment with the MoCo-v2 and MAE models that are pre-trained on medical image datasets. Furthermore, we also experiment with MoCo-v2 and MAE, first pre-trained on medical images and then finetuned on single-view NIH Chest X-ray data. We refer to these fine-tuned models as MoCov2-FT and MAE-FT, respectively.

Since multi-view labeled data can be hard to acquire, we investigated whether it is beneficial to fine-tune the self-supervised pre-trained models on single-view classification before fine-tuning them further for multi-view classification. Comparing the results of MAE with MAE-FT and MoCo-v2 with MoCov2-FT, we clearly observe that additional fine-tuning on single-view classification is greatly beneficial and significantly improves the multi-view classification performance.

Table 4: Mean AUC (across 14 conditions) of models fine-tuned on 1%, 10%, and 100% fractions of the multi-view labeled data. 10 different methods are compared against a random initialization baseline. The pre-training methods generally outperform the random baseline, with the largest gains on low data regime (1% and 10% data fractions).

| Domain | Type | Method | AUC (1%) | AUC (10%) | AUC (100%) |
|--------|------|--------|----------|-----------|------------|
| - | - | Scratch | 60.7 ± 1.02 | 67.1 ± 0.43 | 81.2 ± 0.837 |
| Natural | Supervised | ImageNet | 60.4 ± 3.43 | 70.9 ± 0.95 | 81.5 ± 0.022 |
| | Self-supervised | MoCo v2 | 65.4 ± 4.11 | 69.2 ± 0.28 | 81.3 ± 0.473 |
| | | SimCLR v2 | 65.3 ± 1.67 | 69.4 ± 1.60 | 81.3 ± 0.381 |
| | | Barlow Twins | 47.8 ± 2.14 | 47.9 ± 1.80 | 79.6 ± 0.441 |
| | | BYOL | 56.0 ± 0.51 | 59.0 ± 0.04 | 77.9 ± 0.841 |
| Medical | Supervised | ChestX-ray14 | **70.4 ± 0.07** | **76.7 ± 1.40** | 84.1 ± 0.026 |
| | Self-supervised | MAE | 61.1 ± 4.03 | 68.4 ± 1.52 | 83.5 ± 0.100 |
| | | MoCo-v2 -Med | 70.0 ± 1.24 | 71.3 ± 1.04 | 82.6 ± 1.194 |
| | | REMEDIS | 69.0 ± 3.67 | 73.7 ± 1.26 | **84.3 ± 2.39** |

As presented in Table 3, the results show the benefits of fine-tuning on medical data. Both MoCov2-FT and MAE-FT outperform their respective base models (MoCo-v2 and MAE) across all AUC metrics.

## Appendix C. Detailed Results Across 14 Clinically Relevant Diseases

Here, we investigate the clinical efficacy of the aforementioned pre-training methods by performing a fine-grained analysis on the 14 common diseases covered in ChestX-ray14. In contrast to the 64 conditions studied earlier, some of which might not be very clinically relevant (e.g. electrical device), Table 4 and Fig. 4 compare the AUC scores when evaluated on these 14 clinically relevant diseases. In addition, we also compare the results for these 14 clinically relevant diseases with state-of-the-art methods, as detailed in Table 5.

For reference, the 14 clinically relevant diseases are: Atelectasis, Cardiomegaly, Consolidation, Emphysema, Hernia, Infiltrates, Mass, Nodule, Pleural Effusion, Pleural Thickening, Pneumonia, Pneumothorax, Pulmonary Edema, and Pulmonary Fibrosis.

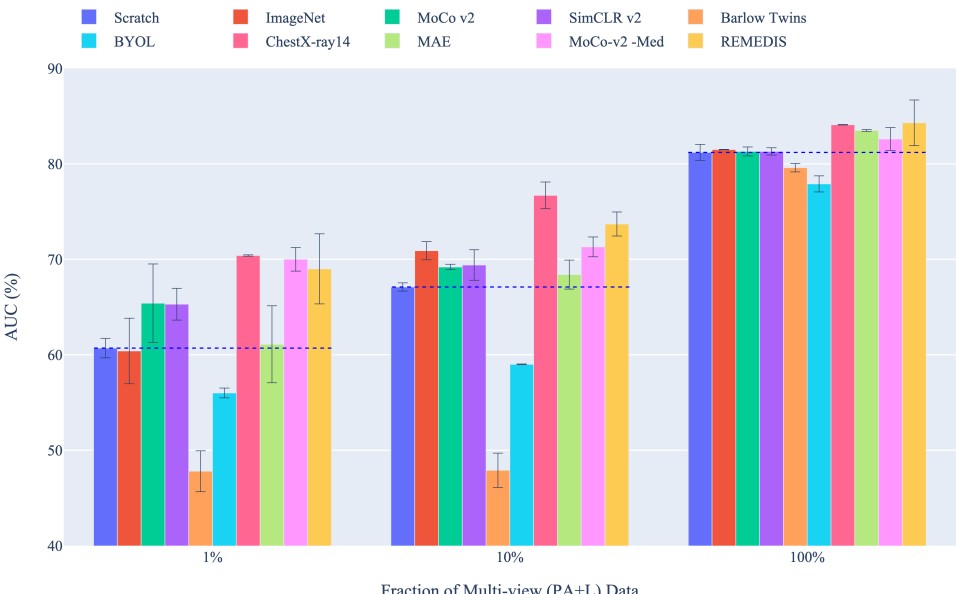

Figure 4: Mean AUC (across 14 clinically relevant conditions) of models fine-tuned on 1%, 10% and 100% fractions of the multi-view labeled data. 10 different models are evaluated including 5 pre-trained on natural images and 4 pre-trained on medical images which are compared against a random initialization baseline. The pre-training methods consistently outperform the baseline, with the largest gains on the data-efficient settings.

Table 5: Comparison with SOTA multi-view chest X-ray classification algorithms across three labeled fractions (for clinically relevant 14 conditions). Our best pre-trained model outperforms the SOTA methods by significant margins.

| Method | AUC (1%) | AUC (10%) | AUC (100%) |
|---|---|---|---|
| DualNet | 52.4 | 64.5 | 82.8 |
| Hemis | 50.4 | 66.1 | 82.6 |
| Stacked | 56.7 | 63.0 | 83.3 |
| AuxLoss | 56.1 | 62.9 | 82.4 |
| REMEDIS | **69.0 $\pm$ 3.67** | **73.7 $\pm$ 1.26** | **84.3 $\pm$ 2.39** |

