# OpenReview forum: "A Comprehensive Benchmark of Supervised and Self-supervised Pre-training on Multi-view Chest X-ray Classification"
_MIDL.io/2024/Conference — MIDL 2024 Poster_

### Official Review · Reviewer_X5ez · 2024-02-24

**Confidence:** 5
**Preliminary Rating:** 1
**Final Rating:** 3.5

**Summary:**

This paper explores the impact of using pre-trained backbone networks on multi-view chest X-ray attribute prediction models.

**Strengths:**

They evaluate multiple self-supervised backbone training methods and compare this to using the weights of a purely supervised model.

Evaluating the improvements in low data regimes using pre-trained models in multi-view is an interesting research question and would be interesting to the community. Guiding the community to use a specific backbone model would speed up many projects.

**Weaknesses:**

If I understand if correctly, in Table 1, the variance shown here is "the mean AUC (on 64 pathologies)" meaning the sum of all individual AUC scores. I do not think this provides the correct variance to compare these models. Having models trained on many different subsets of data and comparing the mean of the mean AUC values of these models would be useful to compare. Using only the mean AUC of one training invocation does not allow us to calculate the expected performance and variance of training that model. So I believe we cannot conclude if one model outperforms another with the current experiments of this paper.

**Detailed Comments:**

It would also be interesting if there were trends for specific pathology performances like pneumonia or effusion between the models.

**Justification Of Final Rating:**

My initial misunderstandings are resolved. I would have also preferred p-values be calculated to assert significance. The paper is likely relevant and interesting to readers. It also involves enough related work and baselines to be a solid contribution.

**Justification Of The Preliminary Rating:**

The experiments performed do not support the statements about performance improvements. If these experiments are redone with another structure, such as taking the mean performance using multiple subsets of data, I would likely change my rating.

**Questions To Address In The Rebuttal:**

Mainly the issue about conclusions about model performance discussed in the previous section.

When training why wasn't early stopping used? Isn't there a concern about overfitting?

---

> ### Author Response · Authors · 2024-03-18
>
> We sincerely appreciate your recognition of our work as “an interesting research question” and the potential benefit of “guiding the community to use a specific backbone model”.
>
> Addressing your concerns:
>
>
> **Clarifying Experimental Setup:**
>
> It seems there has been a misunderstanding regarding our experimental setup described in Table 1. To clarify, as per our Section 3.1: “We conduct each experiment three times, with each run representing a different randomly selected training split to obtain more accurate results.” This approach is precisely what you recommended – “Having models trained on many different subsets of data and comparing the mean of the mean AUC values of these models.”
>
> We apologize for any confusion caused by our initial presentation and agree that this critical detail was not emphasized sufficiently in our paper. In response, we commit to clearly detailing this aspect in the methods section of our revised manuscript, ensuring our experimental design's transparency and rigor.
>
> **Early Stopping/Overfitting**
>
> Our methodology, although not explicitly termed as "early stopping," effectively embodies its principle. We monitored the model's performance on the validation set across epochs and selected the epoch that yielded the best validation results for final evaluation. This strategy aimed to prevent overfitting by identifying the most generalizable point in the training process, akin to early stopping which halts training when validation performance no longer improves.
> We realize our initial manuscript might not have clearly communicated this approach's preventive measure against overfitting. In the revised paper, we will provide a more detailed explanation of this selection process.
>
> We appreciate your feedback and look forward to enhancing our paper with these clarifications.

---

> > ### Comment · Reviewer_X5ez · 2024-03-26
> > **Thanks for the response**
> >
> > I'm not sure how I missed the details about the training splits. Your responses resolve my issues.

---

### Official Review · Reviewer_7mBr · 2024-03-03

**Confidence:** 2
**Preliminary Rating:** 3
**Recommendation:** Poster
**Final Rating:** 4

**Summary:**

This paper presents a comprehensive benchmark study for supervised and self-supervised pre-training methods for multi-view chest X-ray classification. The study evaluates 10 different pre-training models on the PadChest dataset across various data fractions (1%, 10%, and 100%) to understand their performance in multi-view chest X-ray classification. The key findings of this paper lie in the importance of pre-training for data-efficient multi-view classification models. In fact, the study reveals that leveraging larger, unlabeled datasets for pre-training can lead to substantial improvements in model performance on medical imaging tasks. Among the evaluated models, certain architectures demonstrate superior capability to generalize from limited data, suggesting the potential for these models to be deployed in clinical settings where data scarcity is common.

**Strengths:**

The strength of this paper particularly lies in its rigorous benchmarking. It comprehensively evaluates various pre-training models for chest X-ray classification, especially focusing on multi-view images in low data regime.  It provides significant insights into the effectiveness of self-supervised learning methods in medical imaging, highlighting improvements in model performance with limited labeled data. The study is also performed on a large range of clinically relevant diseases.

**Weaknesses:**

While this comprehensive benchmark showcases the best training strategy for multi-view chest x-ray in low data regime, the main limitation of this paper is its focus on a single dataset. In fact, the paper doesn't discuss the potential for dataset-specific biases that may not generalize across different clinical settings or populations. Additionally, while it explores the effectiveness of various pre-training methods, particularly in low-data regimes, the scalability and applicability of these methods to larger, more heterogeneous datasets remain to be fully assessed. Further research is needed to explore these models' performance across diverse medical imaging tasks and settings.

**Detailed Comments:**

The paper has great strength in trying to do a thorough evaluation of multiple transfer learning strategies to provide guidelines on how to perform robust chest X-ray analysis in a low-data regime.
However, the fact that the paper only focuses on a single dataset makes the findings of the study a bit incomplete. How does each strategy perform on other datasets? Is there dataset-specific bias? Is the best transfer learning strategy obtained still robust in other settings?
Other minor improvements:
- Can you give more details about the division of the dataset for generating the 1% and 10% fractions? Do you keep the same population distribution among the fraction or is the division random?
- How is hyper parameter tuning done for the experiments showcased in the study? How do you ensure that all models' results are directly comparable, do you perform hyperparameter tuning for all or none?
- Table 3 in the appendix doesn't show standard deviation of the other models, please add it as it brings good insight of the models' robustness.

**Justification Of Final Rating:**

The authors appropriately addressed my questions pretty well. I'm looking forward to see the additional clarifications and experiments in the manuscript, specially addressing dataset focus and generalisability.

**Justification Of The Preliminary Rating:**

While the paper provide a solid benchmark of different strategies to deal with multi-view chest X-ray analysis. The strength of the paper lies in its particular focus on low-data regime which poses a real challenge in the clinical applicability of deep learning methods. However, the reliance on a single dataset to back-up all the findings seems a little weak and a more thorough discussion on the strategies' generalizabilty would greatly improve the insights provided by the paper.

**Questions To Address In The Rebuttal:**

The main question to address would be about generalizability, the authors need to provide concrete proof of the latter in their study and discuss all the possible biases induced in their work.

**Special Issue:**

No

---

> ### Author Response · Authors · 2024-03-18
>
> We sincerely appreciate your recognition of our study's "rigorous benchmarking" with a “particular focus on low-data regime which poses a real challenge in the clinical applicability of deep learning methods”
>
> **Addressing Dataset Focus and Generalizability**
>
> We understand the concerns raised regarding our study's primary focus on the PadChest dataset and potential dataset-specific biases. Our selection of PadChest in our benchmarking effort is grounded in its significant strengths: it is one of the most comprehensive datasets available, with extensive annotations and a wide variety of multi-view chest X-ray images. These characteristics make PadChest not only a robust base for our study but also potentially representative of the broader clinical landscape. Its diversity and scale offer a unique opportunity to assess the efficacy of pre-training models in a manner that, we believe, can mirror real-world clinical scenarios to a considerable extent. However, we acknowledge the critical importance of generalizability across different datasets and clinical settings. To address these concerns, we are planning to incorporate additional dataset such as MIMIC-CXR to enhance the robustness of our conclusions.
>
> **Dataset Division Clarification**
>
> In creating the 1% and 10% data fractions, we aimed to maintain a distribution of conditions that mirrors the full dataset as closely as possible. This process involved random sampling of images, with an effort to roughly preserve the pathology distribution across each subset. We recognize that this aspect of our methodology was not sufficiently detailed in our initial manuscript and will rectify this in our revised version, ensuring clarity and transparency.
>
> **Hyperparameter Tuning**
>
> Our approach to hyperparameter tuning was designed to ensure fairness and comparability across all models evaluated. We applied a standardized set of hyperparameters across models to maintain direct comparability of results. This uniform approach, crucial for our study’s integrity, will be elaborated upon in the methods section of our revised manuscript to provide clear insights into our experimental design.
>
> We are dedicated to enhancing our paper based on your valuable feedback, aiming to strengthen the implications and contributions of our work.
>
> Thank you for your thoughtful review.

---

> > ### Comment · Reviewer_7mBr · 2024-03-25
> >
> > I would like to extend my appreciation for your thorough response addressing the concerns raised during the review process. Your commitment to enhancing the quality and impact of your work is evident throughout your detailed explanations.
> >
> > **Addressing Dataset Focus and Generalizability**
> >
> > Regarding the dataset focus and generalizability, your rationale for selecting the PadChest dataset is well-founded, emphasizing its comprehensiveness and potential representation of real-world clinical scenarios. I commend your decision to incorporate additional datasets such as MIMIC-CXR to enhance the generalizability of your findings, ensuring that potential biases introduced by dataset selection are acknowledged and addressed.
> >
> > **Dataset Division Clarification**
> >
> > Your clarification on dataset division procedures is crucial for ensuring transparency and reproducibility, particularly when dealing with small data fractions. The detailed explanation of the sampling methodology in your revised manuscript will undoubtedly strengthen the robustness of your findings, allowing readers to better understand the distribution of conditions across subsets.
> >
> > **Hyperparameter Tuning**
> >
> > Regarding hyperparameter tuning, while your standardized approach aims to ensure fairness and comparability across models, it's important to acknowledge that different models may have varying sensitivities to certain hyperparameters. Thus, while applying the same set of hyperparameters is a step towards consistency, it may not fully guarantee fairness of evaluation across models. Clarifying this aspect in your methods section will provide readers with a nuanced understanding of your experimental design and the factors influencing model performance. This clarification will enhance the interpretability of your results and strengthen the credibility of your findings.

---

### Official Review · Reviewer_88m2 · 2024-03-04

**Confidence:** 4
**Preliminary Rating:** 4
**Recommendation:** Oral
**Final Rating:** 4

**Summary:**

The authors present a benchmark for multi-view chest X-ray classification, considering the impact of model pre-training and training data size. The authors hope to popularize the development of multi-view medical imaging models, and raise awareness to the importance of available training data. Their proposed models outperform existing approaches in literature.

**Strengths:**

The proposed experiments are elegant, and to the point. The problem the authors aim to tackle is an important one and the paper is well-written. The authors show a good understanding of the already established methods in the field and their limitations.

**Weaknesses:**

The results on the impact of pre-training are inconclusive: Apart from the supervised and the MoCov2 approach, it is difficult to tell if a method is in itself superior or if it's the impact of pre-training.

**Detailed Comments:**

The authors claim that their code is publicly available, however I did not find a link to their repository.

**Justification Of Final Rating:**

The authors appropriately addressed all my questions and the questions from other reviewers. They also commit to make significant adjustments in the final, revised version of their manuscript, on that condition, I finalize my rating.

**Justification Of The Preliminary Rating:**

The paper is well-designed and written, addressing an important concern. I believe a more thorough discussion can clear up possible reader concerns, strengthening the impact of the proposed benchmark.

**Questions To Address In The Rebuttal:**

- What is the authors opinion on why Balow Twins and the BYOL method performed worse than random initialization?
- Can the authors comment on why the same methods could not be used for comparing pre-training on medical and natural data? Are the differences significant when comparing the results for the Supervised and the MoCov2 methods that are in both pre-training categories? Especially for the 100% data fraction. Why do the authors assume that the good performance of REMEDIS is contributed only to it being pre-trained on medical data, and not simply the performance of that method?

**Special Issue:**

No

---

> ### Author Response · Authors · 2024-03-18
>
> We thank the reviewer for recognizing that we are tackling "an important concern" with a paper that is "well-designed and written," featuring experiments that are "elegant" and "to the point." Below, we address the detailed comments and questions raised:
>
> **On the Inconclusiveness of Pre-training Impact:**
> We acknowledge the reviewer's observation regarding the perceived inconclusiveness of the impact of pre-training. Our benchmarking study was designed to explore a wide array of pre-training methods, acknowledging that their efficacy might vary in nuanced ways. We emphasize that our primary contribution is the systematic comparison of these approaches in the multi-view chest X-ray classification domain, which has not been extensively studied before. To provide further clarity, we will refine our discussion in the paper, more explicitly articulating our findings and delineating which certain pre-training methods demonstrate superior performance.
>
> **Code Availability:**
> We recognize the critical importance of code availability for ensuring reproducibility. In the spirit of open science, we commit to publicly sharing our code.
>
> **On the Performance of Barlow Twins and BYOL Methods:**
> The underperformance of Barlow Twins and BYOL suggests their optimization strategies might not be fully compatible with the intricacies of multi-view chest X-ray data. Specifically, the Barlow Twins' focus on reducing feature redundancy could remove essential, nuanced details vital for accurate X-ray classification. Such details are often critical for diagnosing from medical images. We hypthesize that the Barlow Twins' aggressive decorrelation could thus detrimentally affect diagnostic performance by omitting subtle yet important features. Simlilarly, BYOL relies on an asymmetrical architecture where the online network predicts the target network's representation. This setup might not effectively capture the nuanced differences between multiple views of chest X-rays, where symmetrical learning from each view could be crucial. We will elaborate on these points and our hypothesis regarding their impact in the revised version of our paper.
>
> **On Comparing Pre-training on Medical and Natural Data:**
> The comparison between pre-training on medical versus natural data is indeed a critical aspect of our study. Our analysis did not overlook the comparative effectiveness of methods across these domains; rather, it highlighted the variability in performance enhancements attributed to the domain-specific characteristics of the pre-training data. For intance, MoCov2 which was evaluated across both domains, our findings suggest that the domain alignment of pre-training data (medical versus natural) plays a significant role in model performance, especially in the nuanced task of multi-view classification. This is observed in the 100% data fraction, where the depth and complexity of the data allow for more pronounced distinctions in method performance. We will clarify this point further, detailing the domain-specific performance insights to ensure a comprehensive understanding of these dynamics.
>
> **Regarding the Assumption of REMEDIS's Performance:**
> We appreciate the opportunity to clarify our position on why we attribute the strong performance of REMEDIS primarily to its pre-training on diverse medical datasets. REMEDIS, standing for "Robust and Efficient MEDical Imaging with Self-supervision," is designed as a comprehensive self-supervised learning framework specifically for medical imaging. Its core strength lies in leveraging a wide array of medical data for pre-training, thereby developing robust foundational models for medical image analysis.
> This method's effectiveness in our context is not merely due to its algorithmic design but is significantly enhanced by its pre-training regimen. Pre-training on a broad spectrum of medical images enables REMEDIS to capture the nuanced variations and patterns inherent to medical data, offering a distinct advantage in tasks within the medical domain. In light of the reviewer's feedback we'll clarify how REMEDIS's extensive pre-training on diverse medical datasets is key to its superior performance, beyond just its methodological design.
>
> We thank the reviewer for the valuable insights and the chance to improve our paper.

---

> > ### Comment · Reviewer_88m2 · 2024-03-20
> > **Response to the authors**
> >
> > Thank you for thoroughly addressing my questions, and the questions of the other reviewers. Have the addressed points been corrected in a revised manuscript? It is difficult to tell how the implemented changes change the quality of the manuscript without the revised version.
> >
> > Regarding the performance of REMEDIS, the authors say that they will "clarify" how the performance of REMEDIS is largely dependent on its pre-training. Will this clarification be in the main text or will it be complimented with baseline comparisons, to other architectures? Unfortunately, describing the importance of pre-training will not be sufficient to show the importance of pre-training.
> >
> > The same argument can be said for the "inconclusiveness of pre-training impact", the authors promise to refine the discussion in the paper to better fit the novel application of the approach, but that does not improve the flaws of the experimental design when assessing the significance of model pre-training.

---

> ### Author Response · Authors · 2024-03-24
>
> We thank the reviewer for their constructive feedback and the opportunity to further refine our manuscript. To further clarify our response, we conduct more experimental results rather than only describing the importance of pre-training.
>
> **REMEDIS and Pre-training Impact: Investigate Thoroughly with More Experiments**
>
> To experimentally show that the performance of REMEDIS is largely due to pre-training on medical data, we show the following results:
>
> | Method    | AUC (1%)       | AUC (10%)      | AUC (100%)     |
> |-----------|----------------|----------------|----------------|
> | Scratch   | 60.7 ± 1.02    | 67.1 ± 0.43    | 81.2 ± 0.837   |
> | SimCLR v2 | 65.3 ± 1.67    | 69.4 ± 1.60    | 81.3 ± 0.381   |
> | SimCLR v2 + Medical-Pretraining (REMEDIS)  | 69.0 ± 3.67    | 73.7 ± 1.26    | 84.3 ± 2.39    |
>
> As per the REMEDIS work [1], they use SimCLR as their base method and further pretrain on various medical datasets. In other words, REMEDIS is essentially SimCLR v2 that is pretrained on medical data. Thus, the table above clearly shows that as we pretrain on medical data, the performance improves significantly. We commit to adding the aforementioned experimental results along with explanations to improve our paper.
>
> Furthermore, we are running additional experiments that more explicitly delineate the benefits of pre-training on medical data. This will allow us to better isolate the effects of pre-training from other variables, providing a clearer demonstration of its significance. In particular, to investigate the importance of pretraining, we experiment with MoCo-v2 [2] and MAE [3] that are first pre-trained on medical images and then finetuned on NIH Chest X-ray data. We refer to these experiments as MoCov2-FT and MAE-FT in our results. Comparing the results of MAE with MAE-FT and MoCov2 with MoCov2-FT from Table 1, our initial results indicate that additional fine-tuning on medical data is greatly beneficial and significantly improves the multi-view classification performance. Note: We commit to adding these additional methods and their experiment results to the revised manuscript.
>
> We are grateful for the reviewer's insights. We believe that the additional experiments and clarifications will significantly enhance our work.
>
>
> **References**
>
> [1] Shekoofeh Azizi, Laura Culp, Jan Freyberg, Basil Mustafa, Sebastien Baur, Simon Korn- blith, Ting Chen, Patricia MacWilliams, S Sara Mahdavi, Ellery Wulczyn, et al. Robust and efficient medical imaging with self-supervision. arXiv preprint arXiv:2205.09723, 2022.
>
> [2] Xinlei Chen, Haoqi Fan, Ross Girshick, and Kaiming He. Improved baselines with momen- tum contrastive learning. arXiv preprint arXiv:2003.04297, 2020c
>
> [3] He, K., Chen, X., Xie, S., Li, Y., Doll ́ar, P., Girshick, R.: Masked autoencoders are scalable vision learners. In: Proceedings of the IEEE/CVF Conference on Com- puter Vision and Pattern Recognition. pp. 16000–16009 (2022)

---

> > ### Comment · Reviewer_88m2 · 2024-03-26
> > **Response**
> >
> > Thank you for the adjustments, looking forward to the revised version!

---

### Comment · Area_Chair_uetP · 2024-03-19
**Discussions**

Dear Reviewers The authors have submitted their rebuttal addressing the raised questions. The paper remains open for further discussion and engagement.

---

### Meta-Review · Area_Chair_uetP · 2024-04-03

**Recommendation:** Accept (Poster)
**Confidence:** 5

**Metareview:**

All reviewers unanimously agreed to accept the work for presentation at the 2024 MIDL meeting. The authors are requested to revise the camera-ready version in accordance with the feedback provided in the rebuttal.

---

### Decision · Program_Chairs · 2024-04-05

Accept (Poster)